# Evidence for Topological Protection Derived from Six-Flux Composite Fermions

Haoyun Huang [1], Waseem Hussain [1], S. A. Myers[1], L. N. Pfeiffer[2], K. W. West[2], K. W. Baldwin[2] & G. A. Csáthy [1] ✉

The composite fermion theory opened a new chapter in understanding many-body correlations through the formation of emergent particles. The formation of two-flux and four-flux composite fermions is well established. While there are limited data linked to the formation of six-flux composite fermions, topological protection associated with them is conspicuously lacking. Here we report evidence for the formation of a quantized and gapped fractional quantum Hall state at the filling factor $v = 9/11$, which we associate with the formation of six-flux composite fermions. Our result provides evidence for the most intricate composite fermion with six fluxes and expands the already diverse family of highly correlated topological phases with a new member that cannot be characterized by correlations present in other known members. Our observations pave the way towards the study of higher order correlations in the fractional quantum Hall regime.

The fractional quantum Hall state (FQHS) at the Landau level filling factor $v = 1/3$[1,2] is a prototypical example of a strongly correlated topological phase that hosts quasiparticles with fractional charge[3] and fractional statistics[4]. Improvements of the quality of the host material and advances in cryogenic technology led to the observation of an increasing number of FQHSs. A large number of FQHSs form at Landau level filling factors $v$ that belong to two major Jain sequences: $v = n/(2n + 1)$ and $v = n/(4n + 1)$, where $n$ is an integer. Substituting $n$ with 1, 2, 3,..., one obtains the prominent members of the first Jain sequence $v = 1/3, 2/5, 3/7,...$ and that of the second Jain sequence $v = 1/5, 2/9, 3/13,....$ These sequences were discovered in GaAs/AlGaAs[5–11], and were reported in other high quality host materials, such as graphene[12–17], ZnO/MgZnO[18], and AlAs[19].

Jain's composite fermion theory explains these two major sequences of FQHSs by mapping the strongly interacting system of electrons into a system of nearly free particles, the composite fermions (CFs)[20,21]. The success of the CF theory is rooted in the ability to describe the complex correlations of the system with analytical many-body wavefunctions. The construction of these trial wavefunctions relies on the introduction of Jastrow factors which may be interpreted as electrons capturing magnetic flux quanta. FQHSs are then understood as integer quantum Hall states of CFs moving in a magnetic field modified by the Berry phases associated with the attached fluxes and having a spectrum of quantized and degenerate energy levels called Λ-levels. Specifically, the sequence of FQHSs at $v = n/(2n + 1)$ owes its existence to two-flux CFs ($^2$CFs), whereas that at $v = n/(4n + 1)$ to four-flux CFs ($^4$CFs). CFs, the emergent particles of the fractional quantum Hall regime, are thus a resource for topological protection that arise from the correlated motion of flux tubes, or vortices, and the electrons themselves.

According to the CF theory, FQHSs at filling factors of the form $v = n/(6n + 1)$ are a consequence of the most intricate higher order correlations that stem from six-flux CF ($^6$CF)[20]. The FQHSs at $v = 1/7, 2/13, 3/19,...$ are members of this sequence. However, as discussed in detail later in our manuscript, experiments at such filling factors associated with $^6$CFs so far did not find signatures of topological protection in the GaAs/AlGaAs system. To our knowledge, topological protection associated with $^6$CFs was not reported in any other high quality hosts such as graphene, ZnO/MgZnO, or AlAs either. Here we present results of a search for FQHSs associated with $^6$CFs in a region of filling factors related by the $v \leftrightarrow 1\text{-}v$ symmetry[22] to the filling factors of the form $v = n/(6n + 1)$ in a GaAs/AlGaAs sample that belongs to the newest generation of samples of the highest quality[23]. The electron density is $1.01 \times 10^{11}$ cm$^{-2}$, the low temperature mobility $35 \times 10^6$ cm$^2$ V$^{-1}$ s$^{-1}$, and the width of the quantum well is 49 nm.

[1]Department of Physics and Astronomy, Purdue University, West Lafayette, IN 47907, USA. [2]Department of Electrical Engineering, Princeton University, Princeton, NJ 08544, USA. ✉e-mail: gcsathy@purdue.edu

Topological protection is signaled by the presence of a topological invariant. In the commonly used transport measurements, the presence of Hall quantization $R_{xy} = h/fe^2$, where $f$ is a simple fraction, is the indicator for topological protection. In addition, the opening of an energy gap is a necessary condition for topological protection. In transport, this latter property translates to a decreasing longitudinal magnetoresistance $R_{xx}$ with a decreasing temperature $T$ of the activated form: $R_{xx} \sim \exp\{-\Delta/2k_BT\}$. Here $h$ is the Planck constant, $e$ the elementary charge, $k_B$ the Boltzmann constant, and $\Delta$ the magnitude of the energy gap. A quantized Hall resistance and an activated $R_{xx}$, together with a vanishing $R_{xx}$, are indeed the canonical transport signatures of FQHSs.

We now review experimental data available in the $1/7 \leq \nu < 1/5$ range of filling factors where FQHSs associated with [6]CFs are predicted to form. Early transport in the GaAs/AlGaAs system found weak minima in $R_{xx}$ at $\nu = 2/11$[7] and $\nu = 1/7$[8]. More recent transport confirmed local minima in $R_{xx}$ at $\nu = 1/7$, $2/13$[10,24] and at $2/11$[10] and found local minima in the high frequency conductance at $\nu = 1/7$ and $2/11$[25]. But these local minima did not persist to the lowest temperatures attained in those experiments. The transport signatures observed at $\nu = 1/7$, $2/13$, and $2/11$ are clearly different from those of canonical FQHSs; they were interpreted as evidence of FQHSs at finite temperatures which were engulfed by a competing Wigner solid ground state at the lowest temperatures[10–24]. This interpretation, however, left lingering questions. First, during the long history of the field, developing FQHSs exhibited an inflection in the $R_{xy}$ versus $B$ traces. Evidence for such a Hall signature was seen in an early work at $\nu = 1/7$[8] but was not reproduced later. In fact recent transport did not provide any $R_{xy}$ data[10,24]. Second, past improvements in the mobility resulted in developing FQHSs transitioning into fully developed ones, which exhibited Hall quantization and an energy gap. Well-known examples of this behavior occurred at $\nu = 1/3$[1], $1/5$[6–8], $5/2$[5], and $4/11$[26]. In contrast, a significantly larger mobility of the sample in ref. 24, by more than a factor 25 larger than that in early work[8], did not result in opening of an energy gap at either $\nu = 1/7$ or $2/13$. Third, signatures at the prominent members of this sequence were observed in transport only and not in other experiments. Indeed, microwave absorption[27,28], surface acoustic wave propagation[29], and screening efficiency measurements[30] did access the filling factors $\nu = 1/7$, $2/13$, and $2/11$, but no signatures of FQHSs were reported. To conclude, measurements at $\nu = 1/7$, $2/13$, and $2/11$ reveal a complex behavior. The lack of observation of a quantized $R_{xy}$ or at least

of an inflection in the $R_{xy}$ versus $B$ curves in recent high quality samples remains a weak point of a fractional quantum Hall interpretation at filling factors, such as $\nu = 1/7$, $2/13$, or $2/11$. The ground state at these filling factors in macroscopic samples is an insulator, either the Wigner solid or a crystal of CFs. Even though at these filling factors there is limited evidence for incipient quantum correlations at finite temperatures, these correlations so far are insufficient to establish topological protection associated with [6]CFs.

Numerical experiments in the extreme quantum limit reported a similarly complex behavior. Early simulations considered the competition of Laughlin states and the Wigner solid[2,31–38]. The ground state at $\nu = 1/7$ was overwhelmingly the Wigner solid[32–35,38]. However, the Wigner solid and the FQHS are so close in energy that tuning a parameter of the system, such as the short-range part of the electron-electron interaction[36] or the inter-CF interaction[37], induces a phase transition between the two ground states. More recent numerical work considered the competition between the FQHSs with crystals of CFs, correlated electron solids that can assume lower energies than the Wigner solid[39–42]. References 39–41 found the CF crystal more stable at $\nu = 1/7$. A variational Monte Carlo simulation and a density matrix renormalization group calculation could not clearly discriminate between a fractional quantum Hall and a CF crystal ground state and results of an exact diagonalization study were interpreted as evidence for incompressible fractional quantum Hall ground states[42]. However, this latter technique is known to disfavor crystalline phases. These calculations were performed under certain approximations and therefore it remains to be determined which way the delicate energy balance between these competing phases will be tilted under conditions mimicking those of realistic samples. Melting of an insulating ground state into a finite temperature FQHS as the temperature is raised remains a possibility[34,41].

## Results

Figure 1 shows the longitudinal resistance $R_{xx}$ and the Hall resistance $R_{xy}$ as function of the magnetic field $B$ obtained at $T = 7.6$ mK. For these measurements, our sample was mounted into a He-3 immersion cell[43]; the temperature was measured with a carbon thermometer[44] calibrated against a quartz tuning fork He-3 viscometer[43]. The region above $B = 2.095$ T marks the lowest orbital Landau level, whereas the region at lower fields is the second Landau level. The high quality of the sample is highlighted by the observation in the second Landau level of all eight reentrant insulators[45]. Furthermore, we also observe the even-denominator FQHSs at $\nu = 5/2$[5], a clear signature of the $\nu = 2 + 6/13$ FQHS[46], and a prominent $\nu = 7/11$ FQHS[26]. Near $B = 2.332$ T there is a reentrant integer quantum Hall state associated with the Wigner solid[47].

The dominating feature in Fig. 1 is a very wide $\nu = 1$ integer quantum Hall plateau. At higher magnetic fields, near $B = 5.240$ T, we observe the well-known $\nu = 4/5$ FQHS[5]. Other FQHSs are observed at $\nu = 7/9$ and $10/13$, and there is a pronounced local $R_{xx}$ minimum at $\nu = 13/17$. These four filling factors are of the form $\nu = 1 - n/(4n + 1)$, where $n = 1, 2, 3$, and 4. This sequence of filling factors is related to the major Jain sequence $\nu = n/(4n + 1)$ introduced earlier via the $\nu \leftrightarrow 1 - \nu$ particle-hole conjugation[21,22]. Therefore FQHSs at these filling factors originate from the formation of [4]CFs for which the vacuum is the $\nu = 1$ FQHS. As a consequence, these [4]CFs can be thought of as formed via the flux attachment procedure applied to holes, rather than electrons[21,22]. The Fermi sea of the $\nu = 1 - n/(4n + 1)$ sequence at $\nu = 3/4$ is also marked in Fig. 1. Figure 2 provides a magnified view of the region of interest between $B = 4.8$ T to 5.7 T.

Between $\nu = 1$ and $\nu = 4/5$ there is an unfamiliar magnetoransport feature at $B = 5.120$ T. As seen in Fig. 2, an examination of this feature reveals a Hall resistance plateau. This magnetic field corresponds to the filling factor $\nu = 9/11$ and the value of the Hall plateau is quantized at $R_{xy} = 11\,h/9e^2$, to within 0.14%. At the same time, $R_{xx}$ nearly vanishes at $B = 5.120$ T. These transport features are canonical signatures of a

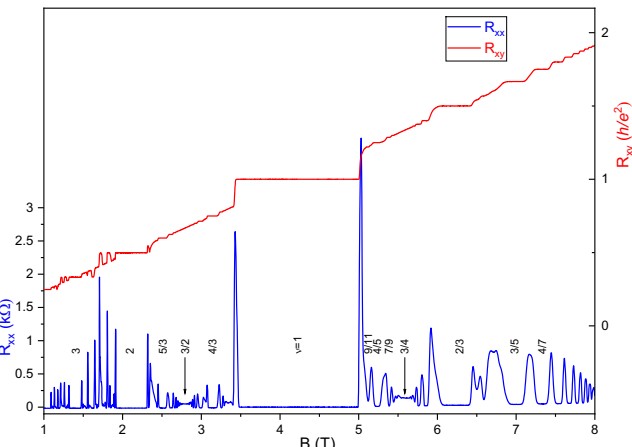

**Fig. 1 | Longitudinal magnetoresistance $R_{xx}$ and Hall resistance $R_{xy}$ as a function of the magnetic field $B$.** Data were collected at the temperature of $T = 7.6$ mK. Several prominent integer and fractional quantum Hall states are marked by their filling factor. In addition, the locations of Fermi seas of CFs are indicated by vertical lines at $\nu = 3/2$ and at $\nu = 3/4$. The region above $B = 2.095$ T corresponds to the lowest orbital Landau level, whereas the region at lower fields is the second Landau level.

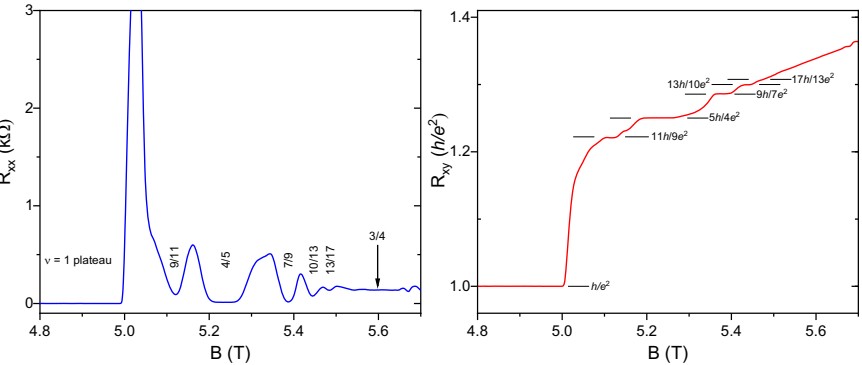

**Fig. 2 | A magnified view of magnetotransport traces $R_{xx}$ and $R_{xy}$ in the range of 4.8 T < $B$ < 5.7 T.** Numbers mark filling factors of prominent FQHSs at $\nu$ = 4/5, 7/9, and 10/13 that belong to the 1-$n$/(4$n$ + 1) sequence, with $\nu$ = 1, 2, and 3. In addition, there is a pronounced local $R_{xx}$ minimum at $\nu$ = 13/17. The most interesting feature of the data is the FQHS at $\nu$ = 9/11 which does not belong to the 1-$n$/(4$n$ + 1) sequence. The vertical arrow indicates the location of a Fermi sea of CFs at $\nu$ = 3/4.

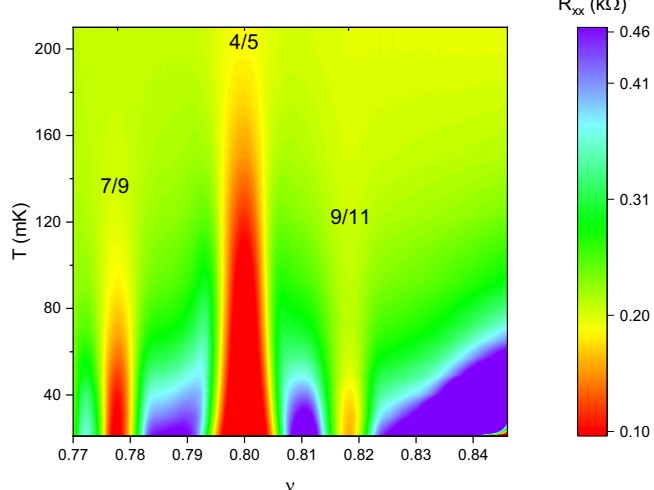

**Fig. 3 | Temperature dependence of the longitudinal magnetoresistance $R_{xx}$ between $\nu$ = 0.77 and 0.845.** Red regions mark areas of low resistance and are therefore indicative of fractional quantum Hall ground states that possess an energy gap in their excitation spectra. The FQHS at $\nu$ = 9/11 is the weakest, albeit fully developed, of the FQHSs observed in this range of filling factors.

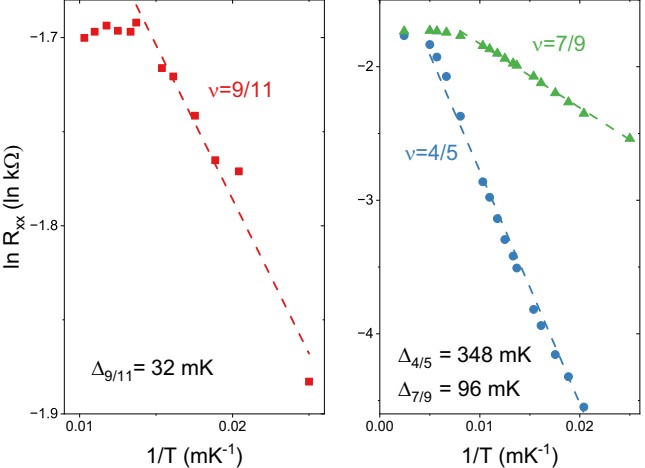

**Fig. 4 | Arrhenius plots of the $R_{xx}$ versus the temperature $T$ at three filling factors of interest.** Energy gaps Δ are obtained from the linear region of these plots at the lowest temperatures from linear fits. Dashed lines show these linear fits to data.

FQHS at $\nu$ = 9/11. Hall quantization is evidence for an edge state dominated transport and thus establishes 9/11 as a topological invariant at this filling factor.

A study of the temperature dependence of our data, shown in Fig. 3, reveals that the FQHS at $\nu$ = 9/11 is one of the most feeble FQHSs we measure. Nonetheless, $R_{xx}$ at $\nu$ = 9/11 decreases with a decreasing temperature. This behavior is shown in Fig. 4. Data for these two figures was obtained after cycling the sample to room temperature; in this cool-down the sample was thermalized through the measurement wires, rather than with He-3. The linear part of the Arrhenius plot shown in Fig. 4 present at the lowest temperatures, i.e. at 1/$T$ higher than 0.0137, establishes that $R_{xx}$ in this region has a thermally activated form $R_{xx}$~exp{-$\Delta_{9/11}$/2$k_B T$}. The lowest temperature data of Fig. 4 therefore provide evidence for the opening of a gap in the energy spectrum at $\nu$ = 9/11. By performing a linear fit, we extract the magnitude of the energy gap $\Delta_{9/11}$ = 32 mK. The presence of an energy gap at $\nu$ = 9/11 indicates incompressibility and ensures an edge-dominated transport. We thus conclude that the observation of a Hall plateau quantized to $R_{xy}$ = 11 $h$/9$e^2$, of an $R_{xx}$ local minimum with a nearly vanishing value, and of the opening of an energy gap provide strong evidence for the formation of a topologically protected FQHS at $\nu$ = 9/11.

## Discussion

We now focus on the origins of the $\nu$ = 9/11 FQHS. As discussed, other nearby FQHSs develop at filling factors of the form $\nu$ = 1-$n$/(4$n$ + 1). The filling factor $\nu$ = 9/11 is not part of either the $\nu$ = 1-$n$/(4$n$ + 1) or the $\nu$ = 1-$n$/(2$n$ + 1) sequences, therefore correlations embodied by $^4$CFs or $^2$CFs cannot account for a FQHS at this filling factor. We conclude that higher order correlations must be at play. In the following we describe two constructions within the CF theory that can explain the formation of this FQHS, both relying on the formation of $^6$CFs. Using the prescriptions of the CF theory, trial wavefunctions can be written down for both constructions[21].

For the first construction, we notice that 9/11 = 1-2/11, the filling factor $\nu$ = 9/11 is thus related via the $\nu$ ↔ 1-$\nu$ particle-hole symmetry to $\nu$ = 2/11[22]. As discussed earlier, at $\nu$ = 2/11 a fully gapped FQHSs has not yet been demonstrated. Nonetheless, the CF theory prescribes a many-body wavefunction for a FQHS at this filling factor. The $\nu$ = 2/11 quantum number belongs to the well-known $n$/(6$n$ + 1) Jain sequence, with $n$ = −2. A negative integer $n$ indicates that the corresponding FQHS develops at a effective magnetic field that points against the direction of the externally applied magnetic field[21]. Within this construction, the $\nu$ = 9/11 FQHS is described as the $\nu^*$=−2 integer quantum Hall effect of $^6$CFs which are built using the flux attachment procedure starting out from holes, rather than electrons. We only invoked $^6$CFs which fill two Λ-levels. Such a FQHS would necessarily be fully spin polarized[21]. We

note that there is a very similar construction for the $\nu = 5/7$ which, however, is based on $^4$CFs. Indeed, the $\nu = 2/7$ FQHS can be understood as the $\nu^* = -2$ integer quantum Hall effect of $^4$CFs and the $\nu = 5/7 = 1-2/7$ FQHS is related by the $\nu \leftrightarrow 1-\nu$ particle-hole symmetry to the $\nu = 2/7$ FQHS.

A second construction is obtained using CFs of mixed flavor. Following steps similar to the ones employed for the FQHS at $\nu = 4/11$[26,48] and at $\nu = 4/5$[49,50], we first apply the flux attachment procedure to generate $^2$CFs. The $\Lambda_2$-level filling factor of these $^2$CFs at $\nu = 9/11$ will be $\nu_2^* = -(1 + 2/7)$. Here the index 2 indicates a description based on $^2$CFs. One may consider the lower fully filled $\Lambda_2$-level inert; a fraction $2/7$ of the second $\Lambda_2$-level is filled. If the $^2$CFs in the upper $\Lambda_2$-level are free, a gap is not expected at the filling factor $2/7$ of this energy level. However, if the $^2$CFs strongly interact, they may capture additional fluxes in order to form CFs of higher order and to condense into the $\Lambda$-levels of these newly formed CFs. The generation of an energy gap through this process is referred to as the fractional quantum Hall effect of composite fermions[26,48]. For the second step of the construction, we apply the flux attachment procedure once more to the $^2$CFs in the topmost partially filled $\Lambda_2$-level, by attaching four more fluxes to each $^2$CFs. The end result of this process is $^6$CFs which will completely fill two $\Lambda_6$-levels of $^6$CFs. Thus, in contrast to the first construction, this second construction for the $\nu = 9/11$ FQHS the formation of both $^2$CFs and $^6$CFs needs to be invoked. The associated trial wavefunctions can account for both a fully spin-polarized and a partially spin-polarized FQHS. However, it is believed that the former is identical to that of the first construction[21]. To summarize, we presented two constructions within the confines of the CF theory to describe the FQHS at $\nu = 9/11$. The topmost $\Lambda$-level for both of these constructions is due to the formation of $^6$CFs, i.e. the valence CFs for both descriptions of the $\nu = 9/11$ FQHS are $^6$CFs.

We now discuss the behavior at other filling factors of the form $\nu = 1-n/(6n + 1)$. The value $n = -1$ yields $\nu = 4/5$. At this filling factor there is a strong FQHS, with an energy gap of $\Delta_{4/5} = 348$ mK. Similarly to the case of $\nu = 9/11$, at $\nu = 4/5$ one could write down a wavefunction based on $^6$CFs filling one $\Lambda_6$ level. However, other wavefunctions of lower order CFs can also be constructed[49,50]. According to the CF theory, dissimilar interpretations are possible and, in such situations, the more natural construction based on CFs of lower order is typically adopted[21]. The value $n = +1$ of the $\nu = 1-n/(6n + 1)$ sequence yields $\nu = 1-1/7 = 6/7$. This value of the filling factor is reached at $B = 4.887$ T. On the $T = 7.6$ mK curves of Figs. 1 and 2, at this value of the $B$-field $R_{xx} = 0$ and $R_{xy} = h/e^2$. Such a transport behavior indicates a ground state with an insulating bulk that is distinct from a FQHS, such as a Wigner solid or a random insulator. Because of the presence of edge states, the former is sometimes referred to as the integer quantum Hall Wigner solid[51,52]. Interestingly, as the temperature is raised, this ground state with an insulating bulk at $\nu = 6/7$ is destroyed. However, in the 7.6 – 400 mK range, in our sample we do not observe a local minimum in $R_{xx}$ at $\nu = 6/7$. Therefore we do not find any evidence for a FQHS at $\nu = 6/7$ even at finite temperatures. We conclude that in our sample only the $\nu = 9/11$ FQHS may be associated with $^6$CFs.

In the regime of formation of $^6$CFs, a Fermi sea is expected in the limit of $n \to \infty$. The parent Fermi sea for the $\nu = 1-n/(6n-1)$ sequence is thus expected to form at $\nu = 5/6$, a filling factor which in our sample is reached at $B = 5.027$ T. Transport at a Fermi sea of CFs exhibits a nearly $B$-field independent and featureless magnetoresistance $R_{xx}$. This behavior may be observed in Fig. 1 for the Fermi seas of CFs forming at $\nu = 3/4$ and $3/2$. As seen in Fig. 2, magnetoresistance at $\nu = 5/6$ is very different. We thus suggest that in our sample a Fermi sea associated with $^6$CFs does not form at $\nu = 5/6$. We think that the unusual magnetoresistance at $\nu = 5/6$ is caused by a competing second ground state, possibly a Wigner solid or a random insulator. While our data is not consistent with a fully developed ground state with an insulating bulk, the integer quantization discussed earlier at the nearby $\nu = 6/7$

associated with a bulk insulator suggests that the same type of bulk insulator also competes at $\nu = 5/6$ with the Fermi sea of $^6$CFs. A similar competition of the Fermi sea with a bulk insulator is known to occur for $^4$CFs at $\nu = 1/4$[1,6,7,18].

In the following we examine other filling factors from the literature at which CFs of mixed flavor, including $^6$CFs, may play a role. A FQHS at $\nu = 6/17$ was proposed to originate from mixed flavor CFs, with one filled $\Lambda_2$-level and another filled $\Lambda_6$-level of $^6$CF[26,48]. Similarly, a FQHS at $\nu = 5/17$ and $\nu = 4/13$ was proposed to be generated by a filled $\Lambda_4$ level and one/two filled $\Lambda_6$ levels, respectively[26,48]. Hall quantization and the opening of an energy gap, thus topological protection, was not yet observed at any of these filling factors[53,54].

We note that the nature of the ground state at $\nu = 9/11$ may change when changing sample parameters. For example, it is known that by increasing the width of the quantum well, a Wigner solid will develop at $\nu = 9/11$ when the electron density is larger than a critical value of about $1.4 \times 10^{11}$ cm$^{-2}$ [51].

In conclusion, we report the observation of an incompressible fractional quantum Hall ground state at $\nu = 9/11$, a Landau level filling factor of the form $\nu = 1-n/(6n + 1)$, with $n = -2$. Hall quantization and the opening of an energy gap at this filling factor indicate a topologically protected ground state with one of the most intricate electronic correlations. Our observations highlight the formation of a new type of emergent fermionic particle, the six-flux CF, a particle that cannot be perturbatively connected to any other CF particles with fewer fluxes.

## Methods
Magnetotransport measurements were performed in a van der Pauw geometry, with an excitation current of 3 nA and employing a standard lock-in technique at 13 Hz. Sometimes the sample state in the GaAs system is prepared by a brief illumination using a red light emitting diode. However, in our experiments we did not employ such an illumination technique, our sample was cooled in dark. The $R_{xx}$ plateaus have a small $B$-field dependent offset. For the energy calculations shown in Fig. 4, these offsets were subtracted.

## Data availability
Data that support the plots within this paper and other findings of this study are available from the corresponding author upon request. Correspondence should be addressed to G.A.C. (gcsathy@purdue.edu).

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

## Acknowledgements

We acknowledge insightful discussions with Jainendra Jain. Measurements at Purdue were supported by the US Department of Energy Basic Energy Sciences Program under the award DE-SC0006671. The sample growth effort of L.N.P., K.W.W. and K.W.B. at Princeton University was supported by the Gordon and Betty Moore Foundation Grant no. GBMF 4420, and the National Science Foundation MRSEC Grant No. DMR-1420541.

## Author contributions

H.H. and G.A.C. conceived the project. L.N.P., K.W.W., and K.W.B. grew the GaAs/AlGaAs wafer and characterized it, S.A.M. fabricated the sample, H.H., W.H. and S.A.M. performed the measurements, H.H., W.H., and G.A.C analyzed the data. The manuscript was written by H.H. and G.A.C. with input from all authors.

## Competing interests

The authors declare no competing interests.
