## [Peer Review File · Nature Communications]

Evidence for Topological Protection Derived from Six-Flux Composite FermionsREVIEWER COMMENTS

Reviewer #1 (Remarks to the Author):

This paper describes the observation, in a 2D system in a GaAs quantum well sample with near record mobility, of a new fractional quantum Hall state, appearing at filling factor $9/11$. The authors argue that the features of this new FQHE state make it likely that this state is comprised of six-flux composite fermions at λ level filling -2 . Unlike prior observations of potential 6-CF states at filling factors $1/7$ and $2/11$, this observation includes a resistance minimum that continues to decrease rather than increase as the temperature is lowered to mK temperatures (7.6 mK in the this experiment). The increase of longitudinal resistance with decreasing temperature observed at $1/7$ and $2/11$, along with DMRG calculations suggests that, for those states, there is a competition with a Wigner Crystal states, either formed by composite fermions or by electrons, and the WC states likely prevail at the lowest temperatures. The authors make a convincing case that mixed CF flavor models, while consistent with observed states at $6/17$, $5/17$, and $4/13$, do not explain the fraction that they observe at $9/11$.

Few laboratories contain the requisite apparatus for these very low temperature measurements, utilizing a helium-3 cell and precision thermometry calibrated against a He-3 viscometer. The GaAs crystals, also, have among the highest mobilities ever measured for electron densities around $1 \times 10^{11} \text{ cm}^{-2}$. I am excited to see what these samples show at fields above those (8T) shown in the data sets in this paper.

I believe that this paper reveals an exciting discovery of a new fractional quantum Hall state along with a credible argument that this fraction arises from novel physics of very complex composite fermions. The paper is very well written, and the arguments are very clear. I recommend publication in Nature Communications.

Reviewer #2 (Remarks to the Author):

The manuscript entitled "Evidence for Topological Protection Derived from Six-Flux Composite Fermions" has reported clear evidence of the $9/11$ fractional quantum Hall state and measured its energy gap. This measurement is from one of the most experienced ultra-low temperature transport groups in the world. Only a small number of labs can provide reliable electron thermometry for such kind of fragile quantum states. The authors also convincingly argue that the $9/11$ state originates from the six-flux composite fermions, which is expected but barely supported in experiments. I think the data is of high quality and should be very valuable if combining information with previous reported fractions.

I have two suggestions. Firstly, I suggest that the authors add a summary of experimentally reported fractional quantum Hall states in their manuscript. The references PRB 77, 075307 (2008) and National Science Review 1, 564 (2014) summarized experimentally reported fractional quantum Hall states. This manuscript probably is an excellent place to update such kind of information. This would strengthen the motivation of the current study and also would be informative to the audience. Secondly, the authors may consider discussing data quality of fractional quantum Hall states reported in monolayer graphene. Researchers frequently make comparisons between fractional states in monolayer GaAs and those in monolayer graphene. The importance of the current work may be better understood in this context.

Reviewer #3 (Remarks to the Author):

In this study, the authors successfully observed the fractional quantum Hall effect at a Landau level filling factor ν of $9/11$. This observation is associated with the formation of six-flux composite fermions. The composite fermion model has succeeded in explaining the sequence of odd denominator FQHEs well by converting the fractional quantum Hall effect originating from a many-body effect of electrons, into the integer quantum Hall effect of composite fermion. Until now, composite fermions with 2-flux CF, which is the most fundamental, and 4-flux CF, observed in higher-quality samples, have been reported. In contrast, this study represents the first reliable example of the fractional quantum Hall effect with 6-flux composite fermions. To achieve this result, the authors grew an ultra-higher-quality GaAs two-dimensional electron system, and evaluated the sample at an extremely low temperature of 7 mK using a specialized refrigerator not commercially available. With these advanced techniques, the authors successfully observed the fractional quantum Hall effect state at $\nu = 9/11$, widely considered one of the most challenging to measure.

This work holds significant interest for the scientific community and thus can be published in Nature Communications. However, I have some questions and comments that the authors should address, which are listed below.

Question 1

When considering only the fractional quantum Hall effect (FQHE) and not the Wigner crystal state, Fermi surfaces of 6-flux composite fermions (6CF) are expected to form at $\nu = 1/6$ and $5/6$, respectively. However, based on the results, the state at $\nu = 5/6$ is mostly in the $\nu = 1$ integer quantum Hall state ($\nu = 5/6$ corresponds to 5.013 T). Intuitively, I think that the development of FQHE should be associated with the development of a Fermi surface. Is this not the case? Please clarify why FQHE can develop without the presence of a Fermi surface.

Question 2

The authors proposed the structure of the $9/11$ edge channel in construction-1 that $9/11 = 1 - 2/11$, and the $9/11$ edge state is written as an integer quantum Hall effect due to hole state with $\nu^* = -2$. This explanation seems easy to understand based on the well-known analogy of $\nu = 1$ and $1/3$, but I think the actual picture is different from the case for $\nu = 1$ and $1/3$. This is because $2/11$ is an integer quantum Hall effect due to the hole state of $\nu^* = -2$ on the Fermi surface of $\nu = 1/6$. There is something strange about the fact that the electron edge channel with $\nu = 1$ is subtracted from the hole edge channel with $\nu^* = -2$, leaving the hole edge channel. How are the cases for in $\nu = 1/4$ or $3/4$ fermi surface? If there is, please explain.

Comment 1

While it may be difficult to avoid confusion in this field due to the variety of fractions involved, I believe it would greatly improve the manuscript if the authors provide a table particularly in the sequences of $1 - n/(6n+1)$ and $1 - n/(4n+1)$.

Comment 2

In the experimental results, the magnetic field for the $\nu = 9/11$ state is crucial and described as 5.120 T, which is a key parameter. The electron density calculated from this magnetic field and the filling factor is $1.01 \times 10^{15} \text{ m}^{-2}$, but this value slightly differs from $1.08 \times 10^{15} \text{ m}^{-2}$ the valued in the manuscript. While it is possible for the electron density to change slightly due to factors such as cool-down processes, it is imperative for the reader's understanding and the integrity of the experimental results that the electron density, magnetic field, and filling factor remain consistent.

Comment 3

On page 2, the authors review the current situation of FQHEs around the $\nu = 1/6$ Fermi surface. However, I find that the transition from this section to the presentation of experimental results feels quite lengthy. While the discussion about the dominance of FQHE and the Wigner crystal around $\nu = 1/6$ is informative, it may not be necessary to review into this level of detail in this manuscript. Instead, I believe it would enhance reader comprehension if this section focused on explaining why the authors chose to investigate $\nu = 9/11$ and the relationship between ν and $1 - \nu$.

Comment 4

In Figure 1, the index of the FQHE that the authors want to emphasize is written on the top axis. However, it is difficult to see which part of the experimental results it corresponds to because the data and the axis are too far apart. If the authors want to emphasize a specific FQHE, I think arrows in a different format than $3/2$ or $3/4$ near the data should be drawn.

Dear Reviewers,

We thank you for your effort and for the positive comments supporting publication. In the following we respond to the concerns of Reviewer #2 and #3. In response to these comments, we made significant changes to our manuscript, and we hope we allayed the concerns brought up.

Yours,

Gabor Csathy

Reviewer #2: "Firstly, I suggest that the authors add a summary of experimentally reported fractional quantum Hall states in their manuscript. The references PRB 77, 075307 (2008) and National Science Review 1, 564 (2014) summarized experimentally reported fractional quantum Hall states."

Our response: We made the following changes to our paper. a) For an improved readability of our paper, we expanded the condensed form of a sequence, such as $n/(4n+1)$ by adding specific examples, such as $1/5$, $2/7$, $3/13$. b) We are also now citing the two reviews and one book chapter we found on graphene. **Changes are highlighted and marked A.**

Reviewer #2: "Secondly, the authors may consider discussing data quality of fractional quantum Hall states reported in monolayer graphene. Researchers frequently make comparisons between fractional states in monolayer GaAs and those in monolayer graphene."

Our response: Graphene is indeed one of the highest quality two-dimensional electron systems. In paragraph #1 of the Introduction we discuss and cite relevant references pertaining to the 2CFs and 4CFs. We added a sentence to paragraph #3 of the Introduction on 6CFs in graphene and ZnO. **Changes are highlighted and marked B.**

Reviewer #3, Question 1: "When considering only the fractional quantum Hall effect (FQHE) and not the Wigner crystal state, Fermi surfaces of 6-flux composite fermions (6CF) are expected to form at $\nu = 1/6$ and $5/6$, respectively. However, based on the results, the state at $\nu = 5/6$ is mostly in the $\nu = 1$ integer quantum Hall state ($\nu = 5/6$ corresponds to 5.013 T). Intuitively, I think that the development of FQHE should be associated with the development of a Fermi surface. Is this not the case? Please clarify why FQHE can develop without the presence of a Fermi surface."

Our response: We now added a dedicated new paragraph, #5 in the Results section, dealing with this topic in detail. At $\nu=1/6$ we do not observe a Fermi Sea of CFs because of a competition with an insulating ground state. **Changes are highlighted and marked C.**

Reviewer #3, Question 2: "The authors proposed the structure of the $9/11$ edge channel in construction-1 that $9/11 = 1 - 2/11$, and the $9/11$ edge state is written as an integer quantum Hall effect due to hole state with $\nu^* = -2$. This explanation seems easy to understand based on the well-known analogy of $\nu = 1$ and $1/3$, but I think the actual picture is different from the case for $\nu = 1$ and $1/3$. This is because $2/11$ is an integer quantum Hall effect due to the hole state of $\nu^* = -2$ on the Fermi surface of $\nu = 1/6$. There is something strange about the fact that the electron edge channel with $\nu = 1$ is subtracted from the hole edge channel with $\nu^* = -2$, leaving the hole edge channel. How are the cases for in $\nu = 1/4$ or $3/4$ fermi surface? If there is, please explain."

Our response: The literature deals with a very closely related construction for the $\nu = 2/3 = 1 - 1/3$ FQHS: PRL 64, 220 (1990), PRB 86, 115127 (2012), PRB 78, 235315 (2008). The first experiments probing the edge of the $\nu = 2/3$ FQHS were recently published PRL 130, 076205 (2023). These measurements do support the idea of a combination of $\nu = 1$ and $\nu = 1/3$ edges.

In our paper, we would like to keep focusing on the nature of the ground states, rather than details of the edges. We made two changes: a) we expanded significantly the discussion of particle-hole symmetry in paragraph #2 of the Results section and we are now citing Jain's book, since it contains the specific details for constructing the relevant FQHS and b) we added to paragraph #2 in the Results section a specific example on how particle-hole symmetry works for the related 4CFs, rather than 6CFs, in explaining the formation of the $\nu = 5/7$ FQHS. **Changes are highlighted and marked D.**

Reviewer #3, Comment 1: "While it may be difficult to avoid confusion in this field due to the variety of fractions involved, I believe it would greatly improve the manuscript if the authors provide a table particularly in the sequences of $1 - n/(6n+1)$ and $1 - n/(4n+1)$."

Our response: We made a change to our manuscript: when the sequences are discussed and their equation is introduced, we also write down the specific members for $n = 1, 2, 3$. These changes are in paragraphs #1 and #3 of the Introduction, and in paragraph #2 of the Results section. **Changes are highlighted and marked A.**

Reviewer #3, Comment 2: "In the experimental results, the magnetic field for the $\nu = 9/11$ state is crucial and described as 5.120 T, which is a key parameter. The electron density calculated from this magnetic field and the filling factor is $1.01 \times 10^{15} \text{ m}^{-2}$, but this value slightly differs from $1.08 \times 10^{15} \text{ m}^{-2}$ the valued in the manuscript. While it is possible for the electron density to change slightly due to factors such as cool-down processes, it is imperative for the reader's understanding and the integrity of the experimental results that the electron density, magnetic field, and filling factor remain consistent."

Our response: This was indeed a typo on our part. We have now fixed this mistake. **Changes are highlighted and marked E.**

Reviewer #3, Comment 3: "On page 2, the authors review the current situation of FQHEs around the $\nu = 1/6$ Fermi surface. However, I find that the transition from this section to the presentation of experimental results feels quite lengthy. While the discussion about the dominance of FQHE and the Wigner crystal around $\nu = 1/6$ is informative, it may not be necessary to review into this level of detail in this manuscript. Instead, I believe it would enhance reader comprehension if this section focused on explaining why the authors chose to investigate $\nu = 9/11$ and the relationship between ν and $1-\nu$."

Our response: Works done at $1/7$ and $2/11$ are well known in the community and the theoretical concepts involved at this filling factors are very closely related to those needed at $9/11$. We feel it is important to highlight both experimental and theoretical efforts at these filling factors and would like to keep this part as it is.

Reviewer #3, Comment 4: "In Figure 1, the index of the FQHE that the authors want to emphasize is written on the top axis. However, it is difficult to see which part of the experimental results it corresponds to because the data and the axis are too far apart. If the authors want to emphasize a specific FQHE, I think arrows in a different format than $3/2$ or $3/4$ near the data should be drawn."

Our response: We changed Fig.1. For an easy readability, we moved the numerical values of the filling factors close to the associated magnetoresistance features. **Changes can be seen in Fig.1.**

REVIEWERS' COMMENTS

Reviewer #1 (Remarks to the Author):

I am happy with the responses and changes that the authors have made to all the reviewer comments and believe that the manuscript should move forward to publication in Nature Communications.

Reviewer #2 (Remarks to the Author):

The authors have addressed the comments from my previous referee report. I think the paper is ready for publication.

Reviewer #3 (Remarks to the Author):

The authors have submitted a revised manuscript along with responses to my questions and comments. The work has been improved by the edit. As a result, I now recommend this work for publication in Nature Communications.

*Please check the typo in the revised part C (is expected).

The only request for a change was from Reviewer 3, he/she asked us to fix a typo. That typo is now corrected.

Gabor Csathy